# Consideration of the Variable Contact Geometry in Vibratory Roller Compaction

**Johannes Pistrol** [1,*], **Mario Hager** [1], **Fritz Kopf** [2] and **Dietmar Adam** [1]

1   Institute of Geotechnics, TU Wien, Karlsplatz 13/220-02, A-1040 Vienna, Austria;
    dietmar.adam@tuwien.ac.at (D.A.)
2   FCP-Fritsch, Chiari & Partner ZT GmbH, Marxergase 1b, A-1030 Vienna, Austria; kopf@fcp.at
*   Correspondence: johannes.pistrol@tuwien.ac.at; Tel.: +43-1-5880122127

**Abstract:** Vibratory rollers are mainly used for the near-surface compaction of granular media for a wide variety of construction tasks. In addition to the pronounced depth effect, vibratory rollers have offered the possibility of work-integrated compaction control (intelligent compaction) for decades. State-of-the-art measurement values for intelligent compaction (ICMVs) only take into account, if at all, a constant geometry of the contact area between the drum and soil. Therefore, this paper introduces a comparatively simple mechanical model, which describes the dynamic interaction between the vibrating drum and the underlying soil during compaction to investigate the influence of the changing geometry of the contact area on the motion behavior of the vibrating drum. The model is tested on realistic soil and machine parameters, and the results of the simulation with varying drum contact geometry are compared to a conventional simulation with a fixed contact geometry. The analysis shows that only a consideration of the varying drum contact geometry can map the dynamic interaction between the vibrating drum and soil sufficiently and provide a motion behavior of the drum that is in good accordance with the field measurements.

**Keywords:** soil dynamics; roller compaction; intelligent compaction; nondestructive testing; compaction control





## 1. Introduction

Near-surface soil compaction in earthworks is mainly performed by vibratory rollers, as they have a greater depth effect than static and oscillatory rollers and therefore allow the installation of greater layer thicknesses. The rotation of an unbalance mounted concentrically in the drum axis of the roller provides for its dynamic excitation and causes a predominantly vertically directed load transfer into the soil.

During the entire compaction process, the dynamically excited drum and the soil form an oscillating, interacting system with changing contact conditions. Depending on the tuning of operating or process parameters (travel speed, excitation frequency, amplitude, etc.), design parameters (ratio of static axle load to vibrating mass, eccentricity, mass of unbalance, etc.) and soil stiffness (grain size distribution, grain shape, mineral structure, water content, bedding density, loading history, etc.), the roller thereby operates in different modes of operation [1]. The resulting characteristic motion behavior of the compaction device can be used to derive and calculate measurement values that allow for an assessment of the compaction state of the soil or its load-bearing capacity. Systems for continuous compaction control (ccc) and intelligent compaction (IC) [1–6] make use of this basic idea.

The dynamic motion behavior of the roller drum as a result of the vibratory excitation is a complex and highly nonlinear process. The occurring nonlinearities result primarily from the curved shape of the drum (geometric nonlinearity), from the plastic behavior of the soil (material nonlinearity) and in particular from the aforementioned interaction between the roller and soil (system nonlinearity due to changing contact conditions).

Compaction work usually starts in "Continuous Contact", where there is permanent contact between the drum of the roller and the soil. During the loading phase, the contact area between the drum and soil surface increases steadily due to the curved drum shape, which in turn influences the reaction forces of the soil and changes the motion behavior of the drum. In the course of the unloading phase, the drum may be lifted off the ground, causing the two subsystems to move separately for a short time. This loss of contact occurs only after the dynamic force between the drum and the soil has assumed a maximum soil reaction force. The motion behavior changes significantly when the two subsystems are separated (change in mode of operation). Thus, the system reacts to the increasing compaction and the resulting increasing soil stiffness with additional nonlinearity (loss of contact). If a loss of contact occurs between the drum and the soil, it is well known that several modes of operation can occur, in addition to "partial uplift", especially "double jump", and possibly also "tumbling" and "chaos" [1]. In this paper, the loss of contact is considered exclusively in the "partial uplift" mode of operation. However, all considerations can also be applied analogously to the "double jump" mode of operation.

In the last decades, numerous research studies have been conducted to better understand the interaction between the vibrating drum of a vibratory roller [2,4,7–9] or an oscillating roller [10,11] and the soil to be compacted.

Lumped parameter models [7,8,12] and cone models [1,3] have already been used in the literature to analyze the drum–soil mechanics but only under the assumption of a constant contact geometry between the drum of the vibrating roller and the soil surface.

Dynamic elasto-plastic finite element models have been developed (e.g., [13,14]) to investigate the interaction between a vibrating drum and the soil. Kenneally et al. [15] already highlight the importance of the consideration of the variable contact geometry between the drum and soil in their FE analysis. Most recent studies primarily focus on the application of machine learning or artificial neural networks (ANNs) on intelligent compaction (e.g., [16]).

This paper presents a comparatively simple semi-analytical vibratory roller–soil simulation model to describe the interaction system, characterized by its challenging contact conditions. Previous models consider, if at all, only a constant geometry of the contact area between the drum and soil. However, the width of the contact area increases during the loading phase when the drum penetrates the soil, making the drum contact width a variable parameter. The presented model is able to account for this variability of the drum contact width in the loading phase of vibratory roller compaction.

The semi-analytical model is part of an ongoing research project with the goal to develop a novel ICMV for vibratory rollers that is largely independent of machine (drum geometry and mass ratios) and process parameters (excitation frequency, excitation amplitude, and travel speed) and, at best, influenced solely by the stiffness of the soil.

First, the equations of motion for modeling the drum of the roller and the soil are derived separately by means of the subsystem technique. This is followed by the coupling of the two interacting subsystems via the formulation of contact conditions (contact problem). In this way, the three possible operating phases per revolution of the exciter—loading, unloading and loss of contact—can be mapped.

## 2. Modeling of the Roller Subsystem

The actual compaction tool is the cylindrically curved drum of the roller, in the axis of which an unbalance (exciter) is mounted on a drive shaft which rotates at a specific exciter frequency $f$. Due to the quadratic excitation, the drum, which is considered rigid, undergoes a circular translatory motion, whereby the soil is compacted by a predominantly vertically directed loading.

The drum is connected to the frame of the roller via rubber buffers, modeled as a Kelvin–Voigt element (a linear elastic spring $k_r$ and a viscous dashpot $c_r$ in parallel). The roller subsystem under consideration thus corresponds to a two-mass oscillator with the absolute displacement of the frame $z_2$ and the absolute displacement of the drum $z_1$ as

position coordinates. The initial position of the entire system in static equilibrium is shown in Figure 1 (left).

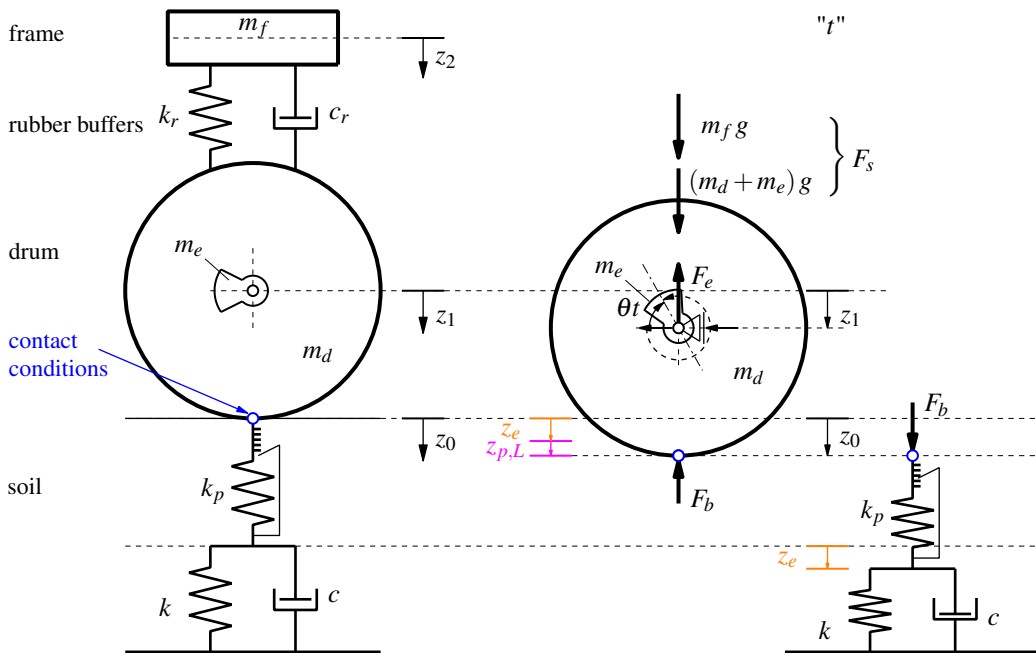

**Figure 1.** Initial position of the entire system in static equilibrium (**left**) and free-body diagram of the simplified simulation model at time $t$ during the first load cycle (**right**).

If the rubber buffers and frame are suitably designed, the frame and drum can be regarded as largely decoupled from vibration so that the energy of the dynamic excitation can theoretically be fully utilized as compaction energy [17]. Under this condition, the frame can be considered to be at rest ($z_2 = 0$), which means that all time-dependent terms of the frame displacement $z_2$ as well as the forces in the rubber buffers $k_r(z_1 - z_2)$ and $c_r(\dot{z}_1 - \dot{z}_2)$ can be neglected [1]. The influence of the frame on the motion behavior of the drum is thus limited to its dead weight ($m_f\,g$), where $m_f$ is the mass of the frame. At the same time, the roller subsystem is now reduced to a single degree of freedom with the position coordinate $z_1$ (see Figure 1, right). From the application of the principle of linear momentum [18],

$$(m_d + m_e)\ddot{z}_1 = -F_b + F_s - F_e, \tag{1}$$

where $m_d$ and $m_e$ are the masses of the drum and the eccentric, respectively, follows the equation of motion for the roller subsystem by means of the resulting soil contact force $F_b$ between the drum and soil:

$$F_b = -(m_d + m_e)\ddot{z}_1 + (m_d + m_e + m_f)g - m_e e\theta^2 \sin(\theta t). \tag{2}$$

The soil contact force $F_b$ in Equations (1) and (2) is defined positive for compression and comprises three parts: the inertia force $F_a$ of the vibrating mass, the static load $F_s$ of the dead weight and the force $F_e$ of the dynamic excitation. $e$ denotes the eccentricity of the unbalance, and $\theta$ is the angular frequency of the excitation. The product of the angular frequency $\theta$ and time $t$ gives the rotation angle of the eccentric mass at time $t$ (see Figure 1, right).

### 3. Modeling of the Soil Subsystem

The soil model presented in the following is not able to represent the processes that occur during dynamic compaction with the vibratory rollers, but it is very well able to describe the compaction state of the homogeneous soil after compaction, which is ultimately also used to assess the compaction quality. It is pointed out that for modeling the dynamic

processes in the soil, the use of higher-order material models (e.g., hypoplasticity [19]) is inevitable. However, the main focus of the presented soil model is on the determination of the resulting soil contact force as well as the governing motion quantities (accelerations, velocities, and displacements). As an essential simplification, only vertical movements on homogeneous soil are taken into account.

The modeling of the soil subsystem is based on considerations by *Wolf* [20]. An arbitrary load application area, in this case, the approximately rectangular contact area of the drum and soil (half side lengths are denoted by $a$ and $b$), is converted into a circle with an equivalent radius, which represents the top surface of a downward unlimited truncated cone and is approximated as a Kelvin–Voigt element. The Poisson's ratio $\nu$ of the soil defines the opening angle of the cone. Vibratory roller compaction is primarily used for cohesionless, compressible soils with Poisson's ratios of $0 \leqslant \nu \leqslant 1/3$, for which *Wolf* [20] does not consider any resonant mass. The associated dynamic parameters of the elastic isotropic half-space, the elastic spring stiffness $k$ and the dashpot coefficient $c$, respectively, are defined as [20]

$$k = \frac{G\,b}{1-\nu}\left[3,1\left(\frac{a}{b}\right)^{0,75} + 1,6\right],\tag{3}$$

$$c = \kappa\,4\sqrt{2\rho\,G\,\frac{1-\nu}{1-2\nu}}\,ab\,...(\text{adapted}),\tag{4}$$

where $G$ is the shear modulus of the soil, and $\rho$ is the density of the soil. The equation for the calculation of the dashpot coefficient presented by *Wolf* [20] is adapted in Equation (4) by introducing the factor $\kappa$. Previous studies [3] have shown that increased damping is required in the one-dimensional model to represent spatial effects and to obtain realistic results.

The essential simplification of the cone model according to *Wolf* [20] can be summarized in the consideration of only one-dimensional waves, which cause a vertical radiation into the soil with constant velocity. The predominantly vertical loading of the soil by a vibratory roller thus justifies the use of *Wolf*'s model [20] for the mechanical model presented in this paper. However, the Kelvin–Voigt element is only able to describe the elastic material behavior like it is observed in the far-field of dynamic compaction work. Therefore, an additional elastic spring with stiffness $k_p$ is introduced, which is connected in series with the Kelvin–Voigt element. The spring deforms proportionally during loading (compression) but remains locked otherwise (during unloading and loss of contact) and, therefore, simulates the near-field of the load application area and enables the consideration of plastic deformations during the loading phase. The elastic half-space remains unaffected. The geometric damping (radiation damping) in the far-field is captured by the damping parameter $c$. Material damping in the near field is not taken into account since it plays only a minor role [21]. Two approaches for the estimation of the spring stiffness $k_p$ are presented in [1]:

Option 1: The stiffness of the spring $k_p$ is estimated by determining the permanent plastic deformation of the soil due to a static roller pass in the field [1]:

$$k_p = \frac{F_s}{z_{p,L}},\tag{5}$$

where $z_{p,L}$ is the accumulative plastic deformation gained during the loading phase, which equals the plastic deformation of the soil after the roller pass. This means that the determined stiffness may well be greater than the stiffness simulating the elastic soil behavior. For well-compacted soils, this will generally be the case.

Option 2: A dimensionless condition factor $\varepsilon$ ($0 \leqslant \varepsilon \leqslant 1$) is defined that describes the ratio of elastic to elasto-plastic deformations and provides a relationship between $k_p$ and $k$ [1]:

$$\varepsilon = \frac{z_e}{z_e + z_p - z_{p,p}} = \frac{z_e}{z_e + z_{p,L}} = \frac{k_p}{k_p + k} \quad \Longleftrightarrow \quad k_p = \frac{\varepsilon\,k}{1-\varepsilon},\tag{6}$$

where $z_e$ and $z_p$ are the absolute elastic and plastic deformations of the soil, and $z_{p,p}$ is a compensation variable to account for the periodic increase in vertical displacements of the soil under cyclic loading, each with respect to the coordinate system of the initial state of the mechanical model. The better the soil is compacted, the lower is the plastic part of the deformations, and the condition factor $\varepsilon$ tends towards a limit value close to 1.

A predefined condition factor thus dictates a certain spring stiffness $k_p$ as a function of the elastic spring stiffness $k_e$ from the cone model. The series connection of the single spring $k_p$ and the elastic Kelvin–Voigt element results in an additional degree of freedom in the soil model (position coordinate of the absolute displacement $z_p$), which is, however, coupled to the absolute displacement of the soil $z_0$.

The soil model presented thus takes into account the fact that deviating conditions from more distant soil zones prevail in the load application area and thus also enable the comparability and separation of relative plastic ($z_{p,L}$) and elastic ($z_e$) deformations. For the calculations and results in this paper, a condition factor of $\varepsilon = 0.85$ is defined to simulate the"partial uplift" mode of operation since this is the desirable mode of operation for vibratory roller compaction.

## 4. Coupling of Roller and Soil Subsystem

The coupling (and separation) of the two subsystems, roller and soil, is characterized by three operating phases—loading, unloading, and loss of contact—which can occur within each period of excitation. Figure 2 illustrates the operating phases on the example of the first period of excitation. The operating phases and the conditions for the transition from one phase to another are explained in the following.

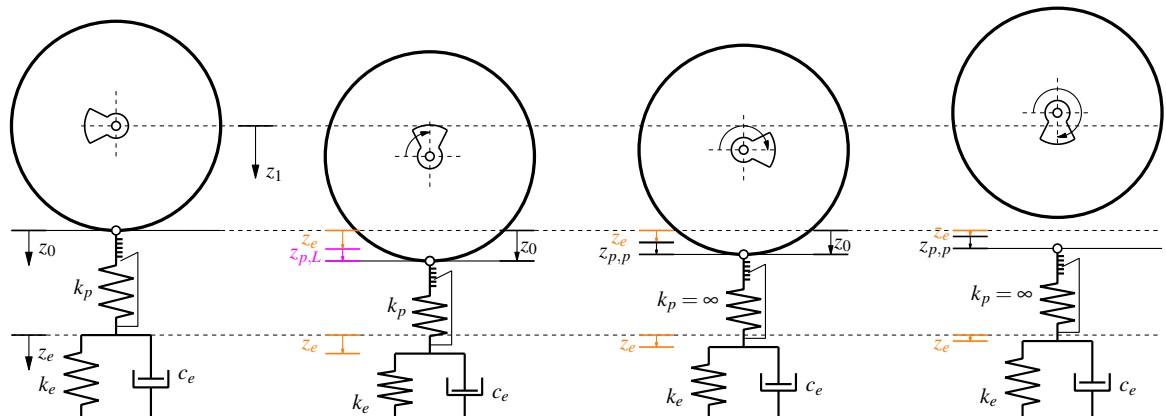

**Figure 2.** Operating phases (1st cycle): initial state, loading, unloading, and loss of contact (f.l.t.r.).

### 4.1. Initial State

For the first increment of the simulation (index $ii = 1$), the initial conditions in the form of the motion quantities of the drum (acceleration $\ddot{z}_1$, velocity $\dot{z}_1$, displacement $z_1$) and the soil ($\ddot{z}_0$, $\dot{z}_0$, $z_0$) are specified or calculated by incremental integration over time, respectively (Equations (7) and (8)). The loading phase is predefined for the following steps of the simulation (indices $ii = 2 - 15$) to ensure a stable beginning of the simulation and to avoid numerical errors.

Initial conditions for the drum:

$$
\begin{aligned}
\ddot{z}_{1(ii=1)} &= \frac{(m_d + m_e + m_f)g - m_e e \theta^2 \sin(\theta t_{(ii=1)}) - F_{b(ii=1)}}{m_d + m_e} = \\
&= \frac{(m_d + m_e + m_f)g}{m_d + m_e}, \\
\dot{z}_{1(1)} &= \ddot{z}_{1(1)} \Delta t, \\
z_{1(1)} &= \dot{z}_{1(1)} \Delta t,
\end{aligned} \tag{7}
$$

where $\Delta t$ is the time step between increments of the simulation.

Initial conditions for the soil are as follows:

$$
\begin{aligned}
\ddot{z}_{0(ii=1)} &= 0, \\
\dot{z}_{0(1)} &= \ddot{z}_{0(1)}\Delta t, \\
z_{0(1)} &= \dot{z}_{0(1)}\Delta t.
\end{aligned}
\tag{8}
$$

### 4.2. Loading Phase

The soil follows the movement of the drum during the loading phase ($z_1 = z_0$). The soil contact force $F_b$, as well as its gradient $\dot{F}_b$, have a positive sign during this operating phase. The spring $k_p$ is active during loading and, therefore, elastic ($z_e$) and plastic ($z_{p,L}$) deformations occur. The sum of the elastic and plastic deformations ($z_e + z_{p,L}$) during the loading phase is the basis for the calculation of the variable drum contact width (see Section 5).

The soil contact force for the roller subsystem is given in Equation (2). The soil contact force for the soil subsystem is the sum of the forces in the elastic spring ($F_k$), and the viscous dashpot ($F_c$):

$$
F_b = F_k + F_c = k\,z_e + c\,\dot{z}_e.
\tag{9}
$$

The subsystems of roller and soil are coupled by the single spring with stiffness $k_p$:

$$
F_b = F_{p,L} = \left[(z_0 - z_e) - z_{p,p}\right]k_p + F_{p,p} = (z_p - z_{p,p})k_p + F_{p,p} = z_{p,L}\,k_p + F_{p,p}.
\tag{10}
$$

The cyclic loading of the single spring with stiffness $k_p$ results in a periodic increase in the (absolute) drum displacement $z_1$, and the soil displacement $z_0$, respectively. Variable $z_{p,p}$ in Equation (10) compensates for this behavior and allows for the calculation of the relative plastic deformations in the loading phase ($z_{p,L} = z_p - z_{p,p}$). Therefore, the compensation variable $z_{p,p}$ describes the cumulative predeformation of the spring $k_p$, which increases by the amount of the plastic deformation $z_{p,L}$ from the previous loading phase at each transition from loading to unloading.

For the "Continuous Contact" mode of operation, in which the drum and the soil remain in contact and an unloading phase is always followed by another loading phase, a residual preload $F_{p,p}$ of the spring $k_p$ must be taken into account. The force $F_{p,p}$ corresponds to the soil contact force $F_b$, with which the spring $k_p$ is "preloaded" at the end of the unloading phase.

The loading phase ends when the soil contact force stops increasing and its gradient is $\dot{F}_b \leqslant 0$. The loading phase is always followed by an unloading phase.

### 4.3. Unloading Phase

In the unloading phase, the soil continues to follow the movement of the drum ($z_0 = z_1$), and Equations (2) and (9) are applicable for the calculation of the soil contact force $F_b$. The gradient of the soil contact force is negative ($\dot{F}_b \leqslant 0$), while the soil contact force itself remains positive ($F_b > 0$). The spring $k_p$ for considering plastic deformations is locked ($k_p = \infty$) and therefore only elastic deformations occur.

The unloading phase may end under two conditions:

Option 1: If the soil contact force $F_b$ becomes zero, the two subsystems separate and move decoupled from each other. The loss of contact phase begins since the soil cannot bear tensile forces and $z_1 < z_0$ applies.

Option 2: If the soil contact force starts to increase again ($\dot{F}_b > 0$) without a full unloading (no loss of contact), the unloading phase is followed by another loading phase. In this case, the residual preloading of the spring $k_p$ is taken into account by the variable $F_{p,p}$.

### 4.4. Loss of Contact Phase

Drum and soil move separately and independently of each other. The absolute displacement of the drum is smaller than that of the soil or, in other words, the drum is above the soil ($z_1 < z_0$) and the soil (assumed to be massless) creeps back towards its initial position during the time of contact loss. The soil contact force in Equations (2) and (9) becomes zero.

The loss of contact phase is always followed by a loading phase as soon as contact between drum and soil is established and the soil contact force is greater than zero. The preload $F_{p,p}$ of spring $k_p$ is set to zero in the case of a loss of contact.

## 5. Variable Drum Contact Width

The contact area between the two subsystems decisively determines the dynamic interaction between the drum and soil. Due to the cylindrically curved drum geometry, the contact area is a variable quantity during the loading phase. The drum increasingly penetrates the soil, and the contact width between the subsystems steadily increases, while the length of the load area remains largely constant and equals the length of the drum.

This circumstance causes significant nonlinearity in the interaction system and thus significantly influences the reaction forces in the soil, which in turn have a decisive influence on the motion behavior of the drum. In contrast to previous research [1,8], the presented model enables the consideration of the variable contact width $2b$ during the loading phase. The instantaneous drum contact width is determined using the circular arc segment shown in Figure 3, which is defined by the sagitta $\Delta z$ and the circular chord $s$.

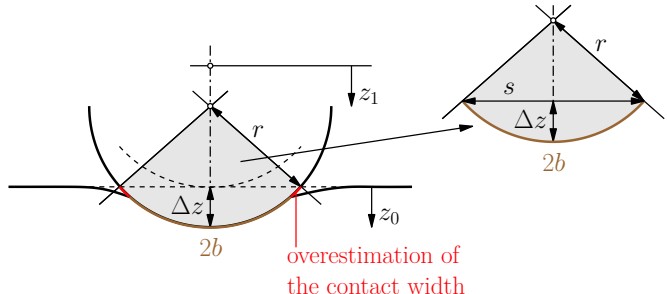

**Figure 3.** Calculation of the variable drum contact width $2b$ during the loading phase.

The sagitta $\Delta z$ is calculated with respect to the relative elasto-plastic deformations of the soil. Therefore, the compensation variable $z_{p,p}$ is subtracted from the absolute soil displacement $z_0$:

$$\Delta z = z_0 - z_{p,p} = ze + z_{p,L}. \tag{11}$$

With the known drum radius $r$ and the calculated elasto-plastic deformations $\Delta z = ze + z_{p,L}$, the circular chord $s$ and, subsequently, the circular arc length $2b$ can be determined according to Equations (12) and (13):

$$s = 2\sqrt{2r\Delta z - \Delta z^2}, \tag{12}$$

$$2b = 2r\arcsin\left(\frac{s}{2r}\right). \tag{13}$$

During the unloading phase, the width $2b$ is assumed to be constant and corresponds to the value at the end of the loading phase since only elastic deformations occur.

If the drum changes directly from the unloading phase to a loading phase without losing contact with the soil beforehand, the contact width has a finite value at the beginning of the loading phase. However, if the drum loses contact and hits the soil again in the subsequent loading phase, an infinitely small contact width would result at the time of contact initialization. This circumstance would inevitably lead to singularities in the further

course of the calculation and consequently to numerical problems in the calculation (see Equations (3), (4) and (11) to (13)). Therefore, a finitely small value for the width of the contact width at the beginning of the loading phase is defined by extending Equation (11) by the variable $\Delta z_T$ in the case of contact loss:

$$\Delta z = z_0 - z_{p,p} + \Delta z_T = ze + z_{p,L} + \Delta z_T. \tag{14}$$

The variable drum contact width results in a variable spring stiffness $k$ and dashpot coefficient $c$ of the Kelvin–Voigt element during the loading phase. A definition of the stiffness of spring $k_p$ via a condition factor $\varepsilon$ (see Equation (6)) also makes $k_p$ dependent on the variable drum contact width. Therefore, an incremental calculation is required.

It is noted that the settlement trough that forms during the loading phase is not taken into account in the presented method for the determination of the contact width and the calculated arc length $2b$ is, therefore, slightly overestimated (see Figure 3). However, in non-cohesive (well-compacted) soils, the influence of the settlement trough plays only a minor role with regard to the drum contact width.

## 6. Calculations and Results

In this chapter, selected results from simulations with the presented mechanical roller–soil model are presented. Table 1 gives a summary of the roller (HAMM H13i [22]) and soil parameters used for the calculations. The soil parameters were selected to ensure the "partial uplift" mode of operation for the roller. To guarantee a steady state of the simulation, 90 revolutions of the eccentric mass were simulated (simulation duration = 3 s). The following graphs show a section of four excitation periods.

**Table 1.** Parameters of the simulation.

| Roller Parameter | Value | Soil Parameter | Value |
|---|---|---|---|
| Diameter of the drum $d$ | 1.504 m | Shear modulus $G$ | 50 MN/m$^2$ |
| Length of the drum $2a$ | 2.14 m | (corresponding) Young's modulus $E$ | 130 MN/m$^2$ |
| Eccentricity $e$ | 23.21 mm | Poisson's ratio $\nu$ | 0.3 |
| Mass of the frame $m_f$ | 3323.0 kg | Density $\rho$ | 1800 kg/m$^3$ |
| Mass of the drum $m_d$ | 3722.53 kg | Contact width $2b$ | variable |
| Mass of the exciter $m_e$ | 69.473 kg | Damping factor $\kappa$ | 4 |
| Excitation frequency $f$ | 30 Hz | Condition factor $\varepsilon$ | 0.85 |

Figure 4 illustrates the development of the variable contact width of the drum during the loading phase (marked "L", highlighted in green). Since the roller operates in the "partial uplift" mode of operation due to the simulation parameters listed in Table 1, the quantity $\Delta z_T$ according to Equation (14) must be taken into account when calculating $\Delta z$ for the loading phase so that the contact area between the drum and the soil assumes a finite value in the first loading increment and singularities are avoided.

In the subsequent unloading phase (marked "U", highlighted in red), the drum contact width is kept constant and thus corresponds to the value at the end of the loading phase. The soil follows the upward movement of the drum until the two subsystems finally separate. During the time of contact loss (marked "T", highlighted in blue), the contact width $2b$ assumes the value zero. In the modeling, however, it must be kept constant (as well as $\Delta z$) in order to determine the soil parameters (see Equations (3) and (4), and Section 5, respectively) via the cone model and to be able to calculate the motion quantities of the soil in the creep-back phase.

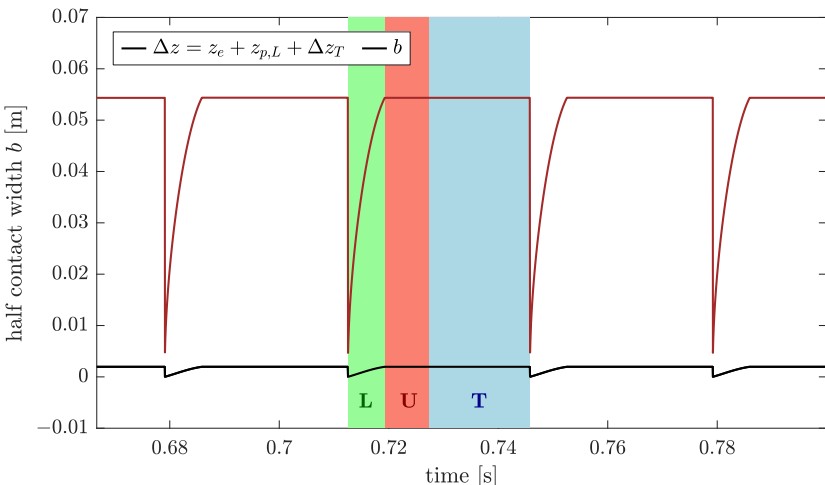

**Figure 4.** Development of the variable drum contact width *b* over four periods of excitation (L ... Loading, U ... Unloading, T ... Loss of Contact) [23].

Figure 5 shows the development of the soil parameters $k$, $c$ and $k_p$ in the individual operating phases (loading, unloading, and loss of contact) resulting from the course of the drum contact width shown in Figure 4. Due to the specification of the condition factor $\varepsilon$ (see Table 1), the spring stiffness $k_p$ also changes during the loading phase (see Equation (6)), which means that all stiffness and damping parameters increase during the loading phase.

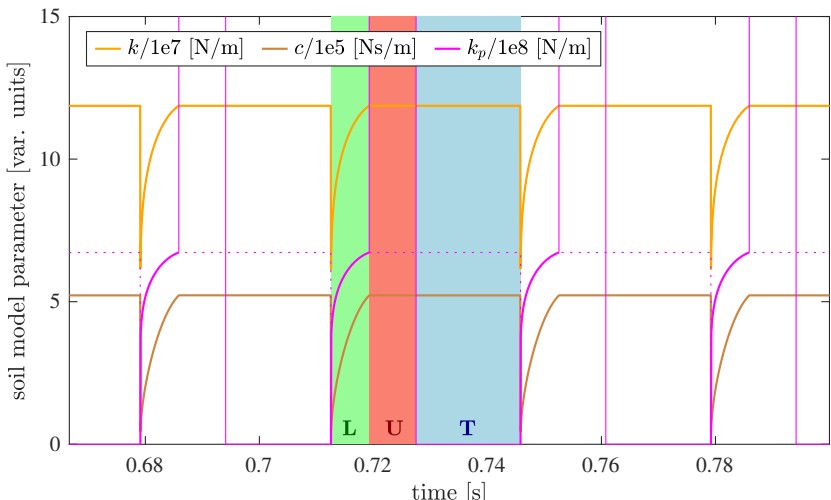

**Figure 5.** Development of soil model parameters $k$, $c$ and $k_p$ depending on the variable drum contact width (L—loading, U—unloading, T—loss of contact) [23].

For the subsequent operating phases unloading and loss of contact, a constant course for the parameters of the Kelvin–Voigt element results due to the given contact width (see Figure 4). According to Equation (6), the state factor $\varepsilon$ in the purely elastically modeled unloading phase approaches its limit value close to 1 and the plastic parameter $k_p$, thus tending to infinity. In the loss of contact phase, on the other hand, the soil provides no resistance. Therefore, the state factor $\varepsilon$ assumes its second limit value and becomes zero, as does the plastic parameter. However, since the spring stiffness $k_p$ is not included in the calculations during unloading and loss of contact anyway, it is kept constant in the modeling, illustrated by the dotted line in magenta in Figure 5.

The determined displacement quantities are shown in Figure 6. According to the described modeling, the roller "moves towards the center of the earth" with constant speed due to the increase in the plastic deformations in each loading phase. In reality, due to its travel motion, the roller constantly moves "uphill" from the impressed deformation

to the soil surface that has not yet been passed over and is still higher at this point in time. The work for this constant "uphill travel" must be conducted by the drive of the roller. Vibration displacement and total deformation are not identical due to the plasticity taken into account. The variable $z_{p,p}$ compensates for this circumstance and allows the calculation of the relative displacements $z_{p,L}$ from the calculated absolute value $z_p$. During the operational phase change from unloading to loss of contact, the drum and the soil separate from each other. In this phase, the soil shows elastic creep back towards its initial position until the drum changes to the next loading cycle and there is again a steady increase in plastic deformations. In the subsequent unloading phase, the soil reacts purely elastically because the single spring is locked ($k_p = \infty$) and thus cannot expand. The maximum of the plastic and elastic deformations does not occur at the same time due to the dynamics in the system (mass inertia and damping).

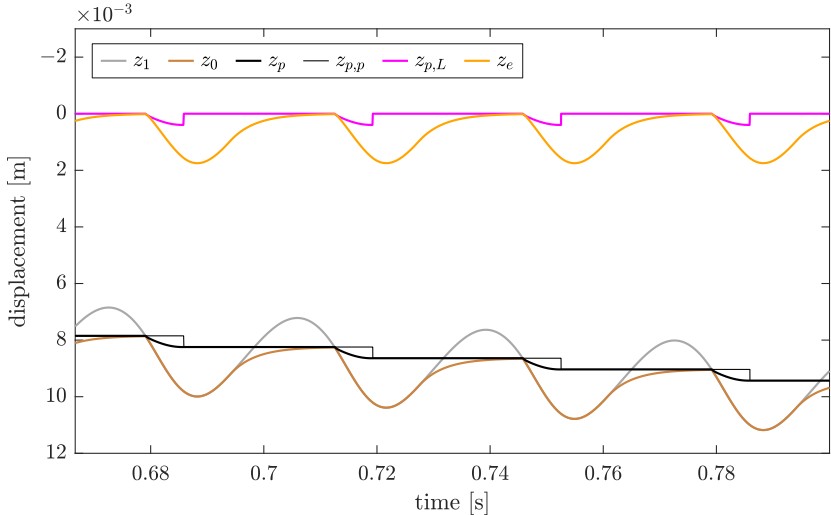

**Figure 6.** Development of displacement quantities over four periods of excitation [23].

Figure 7 shows the temporal course of the determined soil contact force $F_b$ (black) and its components for four excitation periods. The contact force is calculated from its three components, the inertia force $F_a$ of the vibrating mass shifted upwards by the static axle load $F_s$ (blue), and reduced by the force $F_e$ (green) resulting from the rotation of the eccentric mass. If the blue and green lines are superimposed, the soil contact force does not contribute to the inertia force (see Equation (2)). During this period, the drum and the soil subsystems move separately, and the soil contact force becomes zero. A change of the operating phase from loss of contact to loading, or from unloading to loss of contact results in a pronounced kink in the course of the inertia force since the drum acceleration changes abruptly at these points. The inertia force shows recurring movement during each unbalance revolution, amplitudes and contact times are identical, which, in turn, means that the roller operates in the "partial uplift" mode. The maximum contact force coincides with the change from the loading to the unloading phase.

The soil contact force $F_b$ may also be calculated by means of the soil subsystem. The sum of the elastic spring force $F_k$ and the elastic damping force $F_c$ provides the soil reaction force, which becomes zero in the loss of contact phase since the spring and damping forces have the same magnitude. At the moment when the drum reaches the lower reversal point of its motion, i.e., the maximum relative vibration displacement per unbalance revolution, the vibration velocity of the drum and, consequently, also the velocity-proportional damper force $F_c$ become zero.

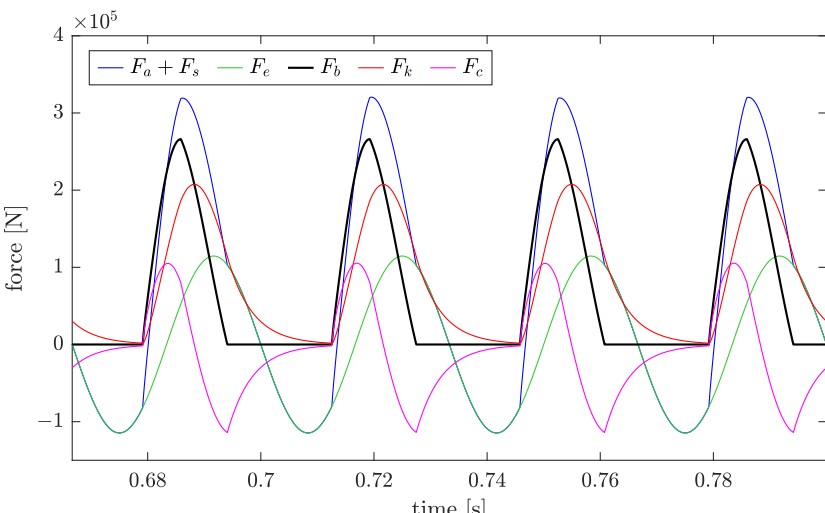

**Figure 7.** Soil contact force $F_b$ and its components for the roller and soil subsystems for four periods of excitation.

Figure 8 shows the course of the plastic spring force $F_{p,L}$ as a function of the absolute plastic deformations $z_p$. These plastic deformations are the cumulative sum of the permanent deformations of each loading phase of the soil (compare Figure 6). The depicted relationship thus illustrates, to a certain extent, the geometric nonlinearity of the system resulting from the consideration of the variable drum contact width and the resulting variable spring stiffness $k_p$ in the loading phase. If the calculations were performed with a constant contact area between the drum and the soil—and therefore a constant stiffness $k_p$—the course of $F_{p,L}$ would be a linear saw-tooth.

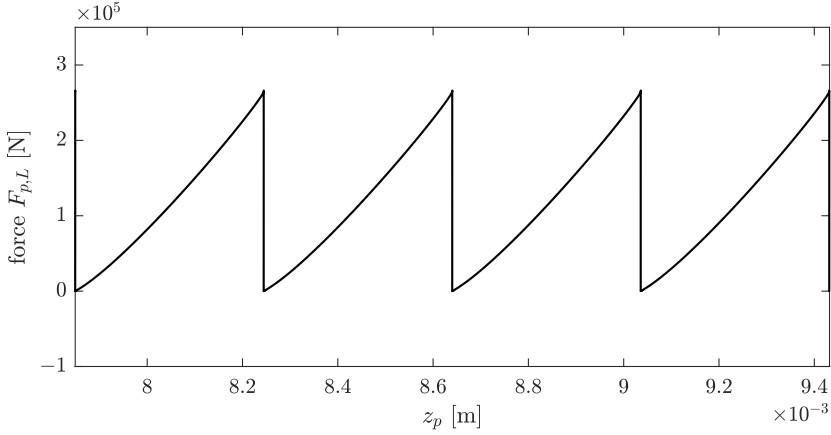

**Figure 8.** Force $F_{p,L}$ in the spring $k_p$ as a function of the plastic deformations $z_p$ for four periods of excitation.

The force–displacement diagram in Figure 9 shows the relation between the soil contact force $F_b$ and the corrected relative vibration displacement of the drum $z_1$ and illustrates the loading and unloading of the soil by the drum. The corrected relative vibration displacement ("folded back" to the initial position of the coordinate $z_1$) is used on the abscissa instead of the total or absolute amount of vertical displacements of the drum in order to ensure the comparability of the diagrams from the simulation with measured force-displacement diagrams.

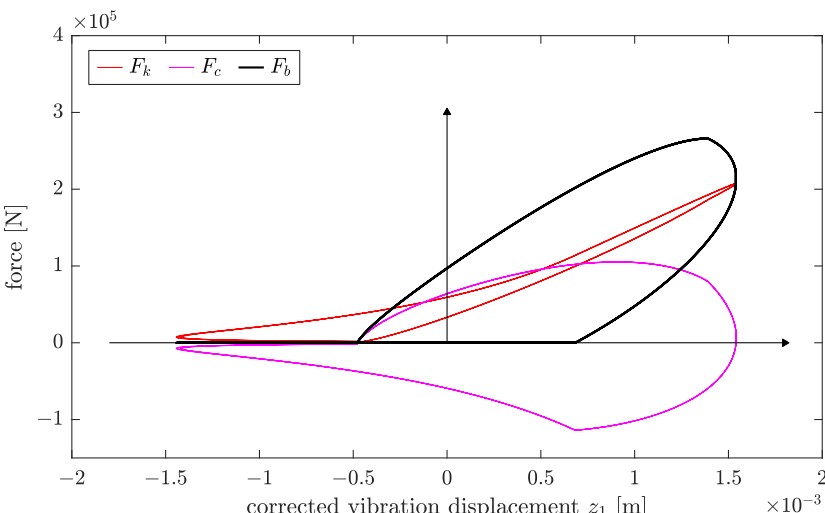

**Figure 9.** Force-displacement curve and its components (soil subsystem) under consideration of the variable drum contact width.

The force-displacement line of the elastic spring force $F_k$ (red) illustrates the main nonlinearities observed during compaction with vibratory rollers: the variable drum contact area—caused by the cylindrical geometry of the drum—and the plastic deformations occurring in the loading phase result in a curved force-displacement line of the elastic spring force during loading. The kink in the soil contact force at the operating phase change from loading to unloading ($F_{b,\max}$) also indicates plastic deformations in the loading phase. Furthermore, the inherent nonlinearity of the system due to the changing contact conditions at the operating phase transitions from unloading to loss of contact, or from loss of contact to loading leads to a kink in the velocity-proportional damper force $F_c$ (magenta).

As can be seen from the determined displacement quantities in Figure 6, the unloading phase represents a purely linear elastic process. Due to the chosen representation of the forces over the corrected vibration displacement of the drum $z_1$, the force-displacement line of the elastic spring force in the corresponding operating phase results in a "loop". This loop disappears for a plot of the elastic spring force over the absolute vertical displacement of the drum. The unloading phase is terminated by the loss of contact ($F_b = 0$); thus, the essential nonlinearity of the system, according to the present contact problem, is captured accordingly. The comparatively simple modeling of the complex interaction system is therefore able to adequately represent the essential nonlinearities and practice-relevant characteristics that occur during dynamic compaction with vibratory rollers.

## 7. Discussion

To further illustrate the relevance of a consideration of the variable contact geometry between the drum and soil, additional simulations were performed with a constant drum contact width. The contact width between the drum and soil was assumed to be $2b = 6$ cm (see Figure 10) and $2b = 12$ cm (see Figure 11), respectively, to ensure comparability with the results in Figure 9. The essential nonlinearity resulting from the curved shape of the drum is thus not taken into account in Figures 10 and 11. The soil parameters $k$, $c$, and $k_p$ therefore assume constant values during the entire simulation.

Compared with Figure 9, the velocity-proportional damper force in Figure 10 shows a much steeper curve at the beginning of the loading phase, while the elastic spring force shows a flatter curve. A comparison of the force-displacement diagrams in Figures 10 and 11, which only differ in the choice of the constant drum contact width, highlights the influence of the predefined contact width on the shape of the force-displacement diagram. The contact area chosen in Figure 11, which is twice as large, already causes an almost abrupt increase in the damper force (and thus also in the soil contact force) at the beginning of the loading phase and has a significant influence on the fullness and shape of the force-

displacement diagram. If the plastic parameter $k_p$ were infinitely large (purely elastic behavior of the soil), or if the spring with stiffness $k_p$ were also connected in parallel with the dashpot $c$, a vertical jump in the damping force $F_c$ (or soil contact force $F_b$) would also result at the transition from the loss of contact phase to the loading phase.

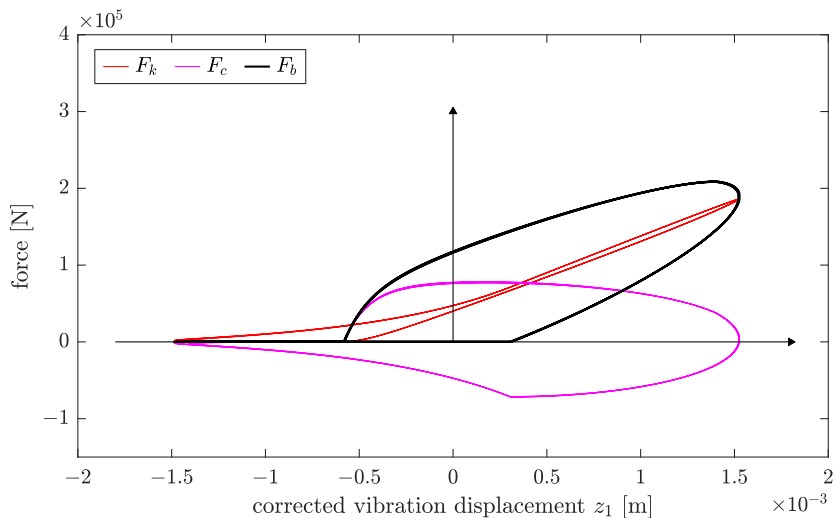

**Figure 10.** Force-displacement curve and its components (soil subsystem) for a constant drum contact width of $2b = 6$ cm.

In reality, however, the area between the drum and the soil at the moment of contact initiation is very small due to the cylindrical curvature of the drum, and thus the contact stress is significant. Therefore, the soil beneath the drum fails in a manner similar to a static ground failure. The impulse due to the drum impact is thus weakened accordingly, and the contact force $F_b$ does not increase abruptly but represents a rather continuous function starting from zero [1].

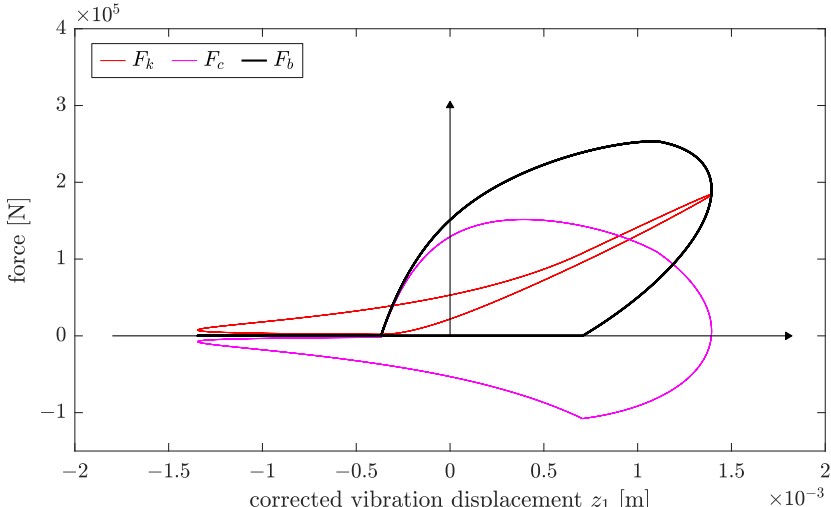

**Figure 11.** Force-displacement curve and its components (soil subsystem) for a constant drum contact width of $2b = 12$ cm.

Force-displacement diagrams with such a shape as shown in Figures 10 and 11 can be observed with good approximation for LWD tests (constant, circular contact area) [21] but do actually not occur for roller compaction in reality [1,17]. Consequently, a consideration of the variable contact width is not only justified but also necessary in order to be able to perform realistic simulations.

## 8. Conclusions

The presented semi-analytical simulation model allows to consider the vertical drum motion (assuming parallel displacement of the drum axis) of a vibratory roller on homogeneous soil, while taking into account the variable drum contact width in the loading phase. In the calculations carried out with realistic roller and soil parameters, the effects of the extended modeling were shown on the example of the "partial uplift" mode of operation. The main findings of the research are as follows:

1.  A comparison of the force-displacement diagrams with and without variability of the drum contact width shows that only the described model extension leads to results that are close to reality and comparable with the measured data.
2.  The simulations with a constant drum contact width show a pronounced overestimation of the damper force after contact initialization. The shape of the resulting force-displacement curves is not in accordance with the measured data from the literature [1,17].
3.  In addition to the system-inherent nonlinearity, i.e., the possibility of the drum lifting off the ground (loss of contact), and the nonlinearity of plastic deformations in the loading phase, the geometric nonlinearity due to the variable drum contact width in the loading phase must consequently also be taken into account.
4.  The "continuous contact" and "double jump" modes of operation, which may also occur during near-surface compaction with vibratory rollers [1], are not considered in this paper but can also be simulated with the presented model.
5.  The model is very well suited for performing efficient parameter studies, as it allows investigations in wide range limits without encountering numerical difficulties and is characterized by short computation times.

The theoretical force-displacement curve in Figure 9 provides the basis for further considerations for the development of a novel ICMV for vibratory roller compaction. In future research, the model will be linked to the in situ measured motion behavior of a vibratory roller to back-calculate the parameters of the soil from the resulting system response (soil reaction based on a force-displacement curve) and thereby derive a novel ICMV.

**Author Contributions:** Conceptualization, J.P. and F.K.; methodology, J.P. and F.K.; software, M.H. and J.P.; validation, M.H. and J.P.; formal analysis, J.P. and M.H.; investigation, M.H. and J.P.; resources, F.K. and D.A.; data curation, F.K. and J.P.; writing—original draft preparation, J.P.; writing—review and editing, J.P. and D.A.; visualization, J.P. and M.H.; supervision, D.A.; project administration, J.P.; funding acquisition, J.P. and D.A. All authors have read and agreed to the published version of the manuscript.

**Funding:** This research was funded by the manufacturer of compaction equipment HAMM AG, Hammstraße 1, 95643 Tirschenreuth, Germany (TU Wien project "794095").

**Data Availability Statement:** The data presented in this study appear in the submitted article.

**Acknowledgments:** The authors acknowledge TU Wien Bibliothek for financial support through its Open Access Funding Programme.

**Conflicts of Interest:** The authors declare no conflicts of interest. The funders had no role in the design of the study; in the collection, analyses, or interpretation of data; in the writing of the manuscript; or in the decision to publish the results.

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
