# Peer review of "Consideration of the Variable Contact Geometry in Vibratory Roller Compaction"

_infrastructures, doi:10.3390/infrastructures8070110_

Round 1

Reviewer 1 Report

The manuscript presents  the studies towards development of the in method for intelligent estimation of soil compaction by the measurement of drum response during compaction process. The importance of considering drum contact loss is highlighted. The paper corresponds to the scope of the journal and could be an important contribution of the research field. Remarks:

1. The  concept of the lCMV is not fully clear: what values are assumes measured on the drum? Is it possible from the drum displacement line z1 in Fig.6 to determine the loss of contact?

2. Fig. 4-7 present a stationary process of homogenous settlement accumulation without deceleration of settlement intensity. Both real soil behaviour and the used Kelvin-Voigt model assume consolidation behaviour.

3. The literature review could be appended with more studies.

Reviewer 2 Report

The authors introducing a mechanical model to describe the dynamic interaction between the vibrating drum and the underlying soil during compaction. And investigating the vertical drum motion of a vibratory roller on a homogeneous soil, while taking into account the variable drum contact width in the loading phase. The manuscript is well writing and other comments are as follows:

1. Line 59, page 2: The author needs to provide a detailed introduction to the concept of "variable drum contact width".

2. More literature review should be included.

3. Figure 1, Page 3: The author did not explain the meaning of "θt" in Figure 1, please add relevant information in the main text.

4. Line 190, page 6: This paragraph needs to be indented in the first line.

5. Page 11: Since the author firstly mention "Figure 8" in line 338, please place the Figure 8 after line 338.

6. The conclusion part should be more concise.

7. Page 14: Please recheck the format of the references. For example, the published year of some references are bolded, while others do not.

Minor editing of English language required.

Round 2

Reviewer 2 Report

The manuscript is now acceptable.

Minor editing of English language required